# Washing Batch Test of Contaminated Sediment: The Case of Augusta Bay (SR, Italy)

**Lucia Lumia [1], Maria Gabriella Giustra [1], Gaspare Viviani [2] and Gaetano Di Bella [1,\***

[1]    Faculty of Engineering and Architecture, Kore University of Enna, Cittadella Universitaria, 94100 Enna, Italy;
       lucia.lumia@unikore.it (L.L.); mariagabriella.giustra@unikore.it (M.G.G.)
[2]    Department of Engineering (DIPING), University of Palermo, Viale delle Scienze, 90100 Palermo, Italy;
       gaspare.viviani@unipa.it
*    Correspondence: gaetano.dibella@unikore.it

**Abstract:** Two experimental campaigns were conducted to optimize the applicability of the Sediment Washing treatment on the marine sediments of Augusta Bay contaminated with heavy metals and total petroleum hydrocarbons (TPH). In the first campaign were used EDTA, citric acid, and acetic acid to removal only heavy metals (Ni, Cu, Zn, Cr, and Hg) from the sediments, while in the second campaign EDTA, citric acid, and EDDS were used to removal heavy metals (Ni, Cu, Cr, and Pb) and TPH. The tests were conducted at different pH values and contact times with 1:10 solid:liquid weight ratio. In the first experimental, at pH values 4, contact time 3 h, and citric acid, high removal efficiencies (78–82%) have been obtained for Ni, Cu, Zn, and Cr metals, while, in the second experimental campaign, at pH value 4, contact time 0.5 h, and citric acid, high removal efficiencies have been achieved especially for Pb and TPH. Finally, on the basis of the results obtained, a conceptual sediment washing treatment layout was proposed and the related costs estimated.

**Keywords:** batch test; dosage; remediation; extraction; sediment washing

---

## 1. Introduction

The Mediterranean Sea is one of the most congested basins in terms of oil tanker traffic, hosting about 20% of the global traffic, and this makes Mediterranean industrial coasts and harbors exposed to severe contaminations [1]. In particular, the Bay of Augusta (Syracuse, Italy is home to numerous industrial and refining activities that have led to a serious state of contamination, both of the water and the underlying sediments, so as to bring the Bay back into the control area of the National Interest Site (SIN) identified by the DM 468/2001. The contaminated Sites of National Interest (SIN) have been defined on the basis of site characteristics, quantity, and hazardousness of pollutants, extent of the environmental impact in terms of health and ecological risk, and the detriment to cultural and environmental heritage. This pollution is mainly due to the presence of toxic compounds, of an organic and inorganic nature, such as polycyclic aromatic hydrocarbons (PAH), polychlorinated biphenyls (PCBs), and heavy metals, but also dioxins, furans, and various pesticides.

Among the most important parameters able to influence the transfer of a pollutant, from the water column to the sediments, are of particular interest [2–4]:

- the physical and chemical characteristics of the pollutants (hydrophobicity, water/octanol partition coefficient, solubility, oxidation state, biodegradability);
- the surface properties of the particles (ion exchange/adsorption capacity, specific surface, organic matter content, buffer capacity); and
- the chemical and physical characteristics of the portion of water in contact with the sediments (pH, Eh, alkalinity, ionic strength, salinity, temperature).

Many of the phenomena and transformations at the base of the transfer from the water column to the sediments are reversible, therefore the sediments can be both the receptor of the contamination and, where contaminated, a potential source of pollution for the aquatic ecosystem [5]. In this context, there are several processing technologies aimed to remove the contaminants present in the sediments and principally aimed to remove the most recalcitrant organic and inorganic contaminants, mainly ex situ. Among the most interesting technologies, thanks to the great managerial flexibility, should be mentioned the extraction and/or washing of sediments. Sediment Washing requires sediment pretreatment aimed at separating the granulometric fractions that compose. Preliminary fractionation reduces the amount of sediment to be treated, which can be limited only to the finest fraction (silt and clay), where most of the pollutants tend to concentrate; on the other hand, the coarser fraction (sand, gravel) may be non-contaminated and, therefore, reusable without treatment.

The studies reported in the literature demonstrate the high potential of the washing techniques, which have often been used satisfactorily for the treatment of contaminated land mainly with heavy metals and hydrocarbons. In particular, the treatment of sediment washing allows one to extract the contaminants from the sediment particles and then remove the soluble pollutants. For this purpose, a wide range of chelating agents can be used, such as ionic and non-ionic surfactants, organic and inorganic acids, but also sodium hydroxide and methanol. The inorganic acids include sulfuric acid ($H_2SO_4$) and nitric acid ($HNO_3$) [6], while the organic acids include citric acid ($C_6H_8O_7$) [7,8] and tartaric acid ($C_4H_6O_6$) [9].

Determining the effectiveness of a chelating agent for washing trace metal contaminated soils has commonly been done in batch extractions at the laboratory scale. Such flask studies shake the soil with a washing solution, usually for 24 h [10]. Washing solutions and contaminated soils have been mixed in large L:S weight ratios of 5:1 [11] and 10:1 [12].

In this context, the study conducted by Khodadoust et al. [13] showed a high efficiency of removal of contaminants such as zinc, lead, and phenanthrene from soils, using as a chelating agent citric acid ($C_6H_8O_7$). Polettini et al. [14] and Lee et al. [15], on the other hand, studied the kinetics of the extraction of heavy metals from sewage sludge, a matrix technically comparable to fine sediments, through the use of citric acid and acetic acid ($C_2H_4O_2$), confirming for both an increase in efficiency with increasing contact time.

Among the most used chelating agents for metal removal, ethylenediaminetetraacetic acid or EDTA and diethylene tramminopentacetic acid (DPTA) are also mentioned [16,17]: numerous studies in the literature show how EDTA is able to remove most heavy metals, such as lead, cadmium, iron, and zinc [18,19].

From Zhuhong et al. [20] a single extraction scheme was proposed with EDTA. This study was carried out by varying different parameters, which is the extraction time, the pH values of the extracting solution and the different granulometric fractions. This study showed that the maximum efficiency of removal on heavy metals is due to the fine fraction and to an extraction time of 24 h. Furthermore, extraction decreases significantly as the pH value increases. Particularly rapid are the kinetics of EDTA complexing with lead [21,22]. Regarding the effect of contact time and of the extracting solid/liquid ratio, Fangueiro et al. [23] conducted a study in which it was found that, for a short time of extraction, it is preferable to use a small amount of sediment, while for higher extraction times, with the same solid-liquid dilution ratio, higher extraction efficiencies are obtained by increasing the sample quantity.

Despite the possibility of obtaining high removal efficiencies, the reduced biodegradability of the extracting agents present could lead to particularly recalcitrant forms of contamination. As a consequence, the identification of further agents with high biodegradability, capable of amplifying the extraction processes, would represent an important and innovative alternative [24]. Most of this kind of research has been carried out in the form of studies comparing the previous EDTA results in metal uptake efficiencies with additional data on the biodegradability of chelants and the metal leaching potential from the application of the chemicals [25]. For this reason, in several studies,

EDDS (ethylenediaminodisuccinic acid) has been proposed as an alternative to EDTA as naturally produced by specific microorganisms.

In this regard, Ritschel et al. [26] conducted some experiments in order to determine the kinetics of metal extraction (Fe, Cd, Cu, Ni, Zn, and Pb) and of the other soil components (Ca and Mn) using EDTA and EDDS as chelating agents. The study was conducted at acidic and neutral pH, in order to evaluate the influence of pH on the extraction kinetics. The results show that the extraction of Fe was more consistent at acidic pH for both chelators. As for Pb, on the other hand, the extraction by EDTA was considerably higher than EDDS in an acidic environment, while the two extractions were comparable to neutral pH.

Wang et al. [27] conducted numerous studies to evaluate the extraction efficiency of Cu, Zn, Cd, and Pb from contaminated soil by EDTA and EDDS at different concentration values. The results obtained shows that EDDS, due to its degradation, was effective only for a certain period of time, depending on its concentration. In particular, high efficiencies for removing Cu, by EDDS, were obtained, while, for other metals, EDTA was more efficient.

On the other hand, Niinae et al. [28] reproduced the previous studies in order to evaluate the applicability of citric acid, comparing the results with those obtained with EDTA and EDDS. The obtained performance data showed clearly better extraction efficiencies with EDTA and EDDS extraction only for Pb, and especially in pH ranges between 7 and 10. According to the study, the results obtained are confirmed by the high values of the constants of dissociation and stability obtained for Pb with EDTA and EDDS, compared to the values obtained for Pb with citric acid. In fact, these constants play a fundamental role in the extraction efficiency of the contaminants, because the higher their value, the more stable the complex and the greater the extraction efficiency. No significant differences were observed with regard to other contaminants.

Indeed, marine sediments are fundamental components of biogeodynamic cycles of mercury and sediments accumulating near industrial and urban areas are reasonably regarded as point sources from which mercury can be mobilized through biological and physico-chemical processes towards surrounding aquatic ecosystems [29]. In sediments of aquatic systems mercury concentrations range between 0.01 and 500 mg kg$^{-1}$. Values lower than 0.1 mg kg$^{-1}$ may be related to natural (unpolluted) areas, while values higher than 1 mg kg$^{-1}$ are generally referred to contaminated areas [30].

The aim of this study was to assess the applicability of the sediment washing treatment for the removal of heavy metals and TPH (C$_{12}$-C$_{40}$) present in the marine sediments dredged from Augusta Bay. Several tests were conducted to determine the influence of pH, contact time, and extraction agents on heavy metals and TPH residual contamination. Finally, the conceptual treatment layout of sediment was proposed, for the implementation of a full-scale plant, and the relative costs were estimated.

## 2. Materials and Methods

### 2.1. Sediment Sampling and Characterization

The sediment were collected from the central and northern Augusta Bay for the first and second experimental campaigns, respectively. The chemico-physical properties of the sediments, as well as their contaminant content, were assessed as follows: The granulometric analysis was assessed by method outlined in ASTM D421-85. Hydrocarbon concentration, expressed as mg kg$_{DW}$$^{-1}$, was assessed by GC-FID (Agilent 6890N, Cernusco sul Naviglio (MI), Italy) as several total petroleum hydrocarbons (TPH) (US EPA 8015C). EPA 3545A and the "Speed Extractor E-916" were used for the extraction procedures. Finally, heavy metals concentration was assessed by ICP AES (OPTIMA 4300DV, Perkin Elmer®) with the EPA 3051A method. "Discovery Cem®" was used for the digestion procedures (EPA 6010C). After collecting, the sediments were stored in the laboratory at 4 °C to prevent biodegradation of the organic matter [14]. Finally, the samples were dried slowly in air for 72 h, homogenized, and sieved. For the first experimental campaign, the analyzes to determine the content of metals and TPH present in the sediment, carried out on the original sediment on the granulometric

fraction between 2 mm and 63 μm and on the granulometric fraction less than 63 μm, while, for the second experimental campaign, only on the granulometric fraction greater than 63 μm and less than 63 μm. This was decided on the basis of consideration made during the first campaign and described in Section 3.1.

## 2.2. Extraction Experiments

The tests conducted during the all experiments were performed with 200 g of sediment and 2 L of buffer solution containing the chelating agents (1:10 S:L weight ratio), under shaking condition (200 rpm). In the first experimental campaign, tests on sediment samples sieved a 2 mm and 63 μm were performed, in order to assess the influence of particle size on treatment performance. The tests were conducted using different chelating agents such as EDTA, citric acid and acetic acid, at different concentrations (0.05 M, 0.1 M, 1 M,), contact times (3 h and 24 h), and pH equal to 4. In the second experimental campaign, tests on sediment samples sieved at 63 μm were performed. On the basis of the results obtained, it was decided to continue the study on metal removal using as agents EDTA, citric acid and EDDS, with a extractant concentration of 0.05 M and deepening the study on contact times (0.5 h and 5 h) and pH (4 and 9). As for the TPH, the tests were carried out with the same extracting agents used for metals at pH 4, 7 and 9 and for the contact times of 0.5 h, 2 h, 4 h, and 5 h. For each test, the residual concentrations of contaminants in the sediment were determined and, consequently, the removal efficiencies. All measures were carried out in triplicate to obtain reliable data and the results presented here represent the average of three value-independent measurements.

## 3. Results

### 3.1. Sediment Characterization

Table 1 shows the chemical-physical characterization of the sediments of the first experimental, sampled in different point ($S_1$, $S_2$, $S_3$, and $S_4$) of the central area of Augusta Bay. The grain size distribution of the material revealed a greater quantity of the sandy fraction (40.120%), followed by silty (37.654%) and clay (22.226%) fractions. The measurements carried out for the determination of the concentrations of heavy metals present in the sediments showed a high contamination of heavy metals and, in particular, Ni, Cu, Zn, Cr, and Hg. With respect to the measurements made on TPH, the results obtained show low values (in each fraction analyzed) and below Italian regulatory limits.

The results obtained from the characterization highlight some interesting aspects concerning the applicability of the intervention treatments and the interpretation of the characterization:

- in the case of "heavy metals", the original sediment shows contaminations that worsen critically once the sediment is pre-fractionated; and
- for almost all samples, the contamination is mainly concentrated in the fine fraction, probably due to the presence of a greater percentage of organic material on the fine fraction (see also TPH data).

In order to improve the consultation of the data in the table, Figure 1 shows the results of the characterization of the metals of each sampling point (S) in the central area of Augusta Bay.

It is interesting that the possible recovery of the coarse fraction, potentially uncontaminated, could reduce the weight and volume by about 30% of the fraction to be treated and/or disposed. On the other hand, there exists a serious risk that, after separation, the pollutants are concentrated in the fine fraction and, consequently, a higher removal efficiency must be required for any treatment of decontamination (or management as hazardous waste). In this sense, the chemical characterization of the untreated sediment would not be exhaustive for the design of ex situ reclamation of any dredged sediments.

**Table 1.** Sediment characterization in the sample of the first survey campaign.

| Parameter | UM | Initial Unfractionated Sediment | | | | |
| --- | --- | --- | --- | --- | --- | --- |
| | | $S_1$ | $S_2$ | $S_3$ | $S_4$ | Composite Sample |
| TPH ($C_{12}$-$C_{40}$) | mg kg$_{DW}$$^{-1}$ | 30.030 | 60.600 | 220.000 | 114.020 | 177.830 |
| Ni | mg kg$_{DW}$$^{-1}$ | 65,905.249 | 45,080.000 | 12,540.000 | 21,813.600 | 55,834.000 |
| Cu | mg kg$_{DW}$$^{-1}$ | 54,821.110 | 48,840.000 | 89,740.000 | 116,331.500 | 103,235.400 |
| Zn | mg kg$_{DW}$$^{-1}$ | 78,173.752 | 6672.020 | 204,233.333 | 187,920.000 | 196,075.012 |
| Cr | mg kg$_{DW}$$^{-1}$ | 118,149.867 | 78,040.001 | 21,780.000 | 101,211.200 | 109,680.500 |
| Hg | mg kg$_{DW}$$^{-1}$ | 83.002 | 11.000 | 1.550 | 45.290 | 27.500 |
| Parameter | UM | Sediment Fraction < 63 µm | | | | |
| | | $S_1$ | $S_2$ | $S_3$ | $S_4$ | Composite Sample |
| TPH ($C_{12}$-$C_{40}$) | mg kg$_{DW}$$^{-1}$ | 30.760 | 70.020 | 420.000 | 172.210 | 250.000 |
| Ni | mg kg$_{DW}$$^{-1}$ | 69,983.966 | 47,767.000 | 12,000.000 | 25,020.900 | 58,491.983 |
| Cu | mg kg$_{DW}$$^{-1}$ | 57,327.293 | 51,010.000 | 82,000.000 | 134,370.100 | 129,500.000 |
| Zn | mg kg$_{DW}$$^{-1}$ | 81,927.991 | 944.400 | 230,043.313 | 258,080.000 | 245,665.670 |
| Cr | mg kg$_{DW}$$^{-1}$ | 125,293.333 | 82,070.000 | 18,000.008 | 140,843.200 | 133,065.000 |
| Hg | mg kg$_{DW}$$^{-1}$ | 6.067 | 5.730 | 8.100 | 32.001 | 15.007 |
| Parameter | UM | Sediment Fraction < 63 Micron | | | | |
| | | $S_1$ | $S_2$ | $S_3$ | $S_4$ | Composite Sample |
| TPH ($C_{12}$-$C_{40}$) | mg kg$_{DW}$$^{-1}$ | 10.000 | 10.430 | 330.000 | 40.000 | 40.230 |
| Ni | mg kg$_{DW}$$^{-1}$ | 19,205.000 | 31,200.000 | 15,205.080 | 15,209.800 | 17,222.700 |
| Cu | mg kg$_{DW}$$^{-1}$ | 25,920.400 | 3400.000 | 125,316.030 | 77,948.300 | 80,141.200 |
| Zn | mg kg$_{DW}$$^{-1}$ | 35,317.000 | 49,400.200 | 71,782.901 | 38,825.600 | 55,397.700 |
| Cr | mg kg$_{DW}$$^{-1}$ | 36,273.200 | 48,803.800 | 39,100.000 | 17,539.000 | 26,653.000 |
| Hg | mg kg$_{DW}$$^{-1}$ | 95.176 | 11.000 | 1.000 | 13.481 | 8.228 |

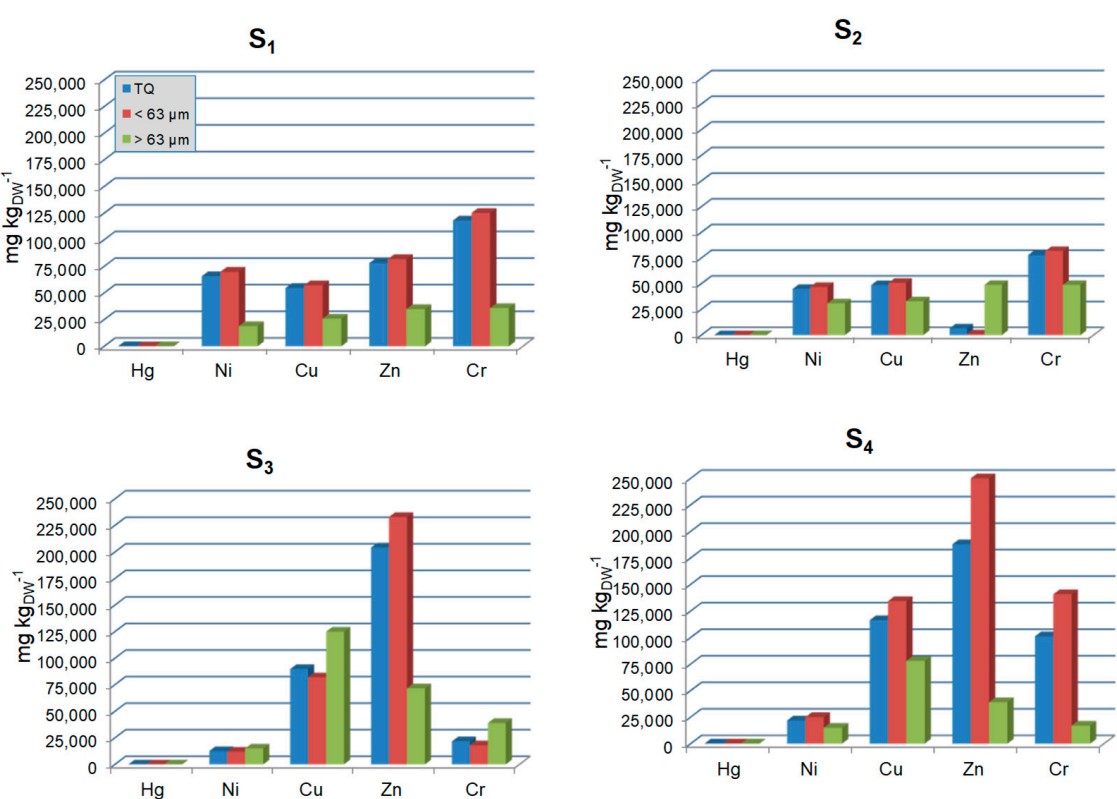

**Figure 1.** Results of the characterization conducted during the first survey campaign at different sampling points ($S_1$–$S_4$, section of the sample).

As for the second experimental campaign (Table 2), the grain size distribution (sand (42.360%), silt (35.380%), and clay (22.270%)) is very similar to the sediment of the first experimental campaign.

The analyses to determine the concentrations of metals and TPH in the sediment were carried out only on the sediment fraction greater than 63 μm and the sediment fraction less than 63 μm. In particular, the results showed high values of Ni and Cu (although lower than in the first campaign) and absence of metals Zn and Hg compared to the previous campaign. Furthermore, Pb was found in these samples.

**Table 2.** Results of the characterization conducted during the second survey campaign.

| Parameter | UM | Composite Sample | | |
|---|---|---|---|---|
| | | **Unfractioned Sediment** | **<63 μm** | **>63 μm** |
| **TPH (C$_{12}$-C$_{40}$)** | mg kg$_{DW}^{-1}$ | 13,567.400 | 17,840.900 | 9561.002 |
| **Ni** | mg kg$_{DW}^{-1}$ | 19,482.100 | 37,430.320 | 500.900 |
| **Cu** | mg kg$_{DW}^{-1}$ | 170.010 | 278.100 | 180.320 |
| **Pb** | mg kg$_{DW}^{-1}$ | 8525.020 | 14,343.510 | 850.010 |
| **Cr** | mg kg$_{DW}^{-1}$ | 11,090.300 | 19,411.150 | 704.310 |
| **Hg** | mg kg$_{DW}^{-1}$ | – | – | – |

Finally, a high TPH contamination equal to 9500.028 mg kg$_{DW}^{-1}$ and 17,800.002 mg kg$_{DW}^{-1}$ were found, respectively, for the sediment fraction greater than 63 μm and the sediment fraction less than 63 μm.

The results obtained from the characterization shows that:

- "inorganic" contamination is concentrated in the fine fraction; and
- in the case of oil pollution, the splitting operation shows a balanced distribution of TPH in both the fine and coarse matrices, with a tendency to concentrate in the fraction less than 63 micrometers.

In summary, the second campaign made it possible to establish a useful study matrix for assessments of decontamination tests aimed at analyzing organic and inorganic pollution.

*3.2. Performance of Heavy Metal Removal with Sediment Washing*

The assessment of potential treatment capacity of sediment washing, in terms of heavy metal removal, was initially performed analyzing the results of the first experimental campaign. In this case, the washing tests were carried out with laboratory batch tests on the fine sediment fraction previously sieved (<63 μm), as it more contaminated. The washing tests were conducted with:

- EDTA, a widely used agent for contaminated sediment applications, which demonstrated good extraction efficiency for many heavy metals;
- citric acid, due to its high biodegradability, being a natural product, deriving from the metabolism of most living organisms;
- acetic acid, a more degradable and less toxic compound than EDTA, but having extractive and lower-cost properties; and
- deionized water, to study metal mobility. The comparison between the different selected chelating agents (Figure 2) shows that the most performing chelating agent, in terms of removal efficiency, is the citric acid. As far as the influence of concentration, it varies according to the chelating agent and the species to be complexed. In general, comparing the performances according to the concentration of the best agent (which is somehow very economical), it can be seen, already, that, at 0.05 M, the efficiencies are more than satisfactory, also on the basis of the reduction in the use of additives.

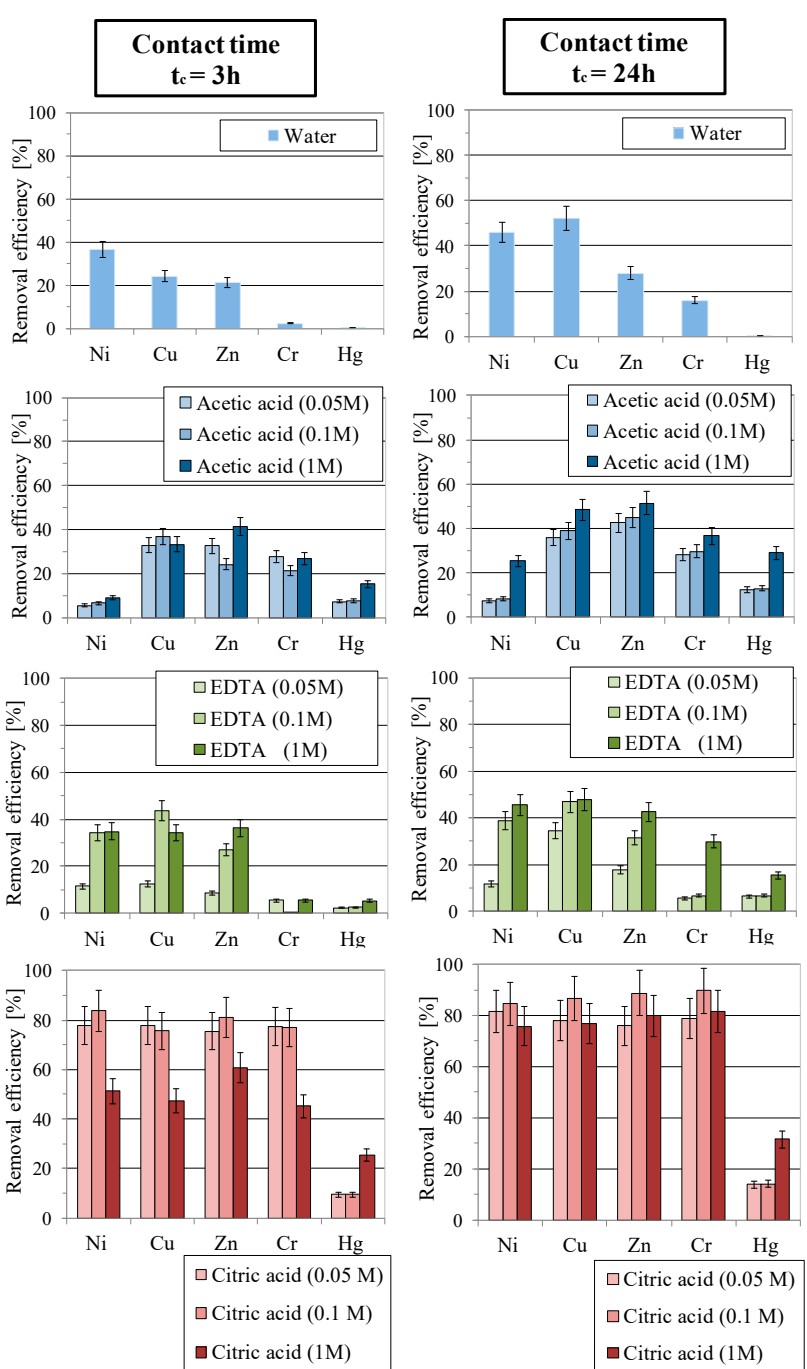

**Figure 2.** Efficiency of removal of inorganic pollutants in the washing tests of the first survey campaign.

On the other hand, the parallel study on contact time showed that as time increases, from 3 h to 24 h, the metal removal efficiencies increase and, in particular, in the case of the 1 M concentration of citric acid, a significant increase in performance was found. With respect to Hg, the increase in concentrations and contact time does not seem to favor a satisfactory decontamination. The highest efficiencies (~37%) were obtained with 1 M citric acid and 24 h contact time.

In order to complete the survey scenarios, Figure 3 shows the most significant removal efficiency values for heavy metals using EDDS (in addition to EDTA. acetic and citric acid) and pH control, based on the tests carried out on the "composite sample" of the second investigation campaign.

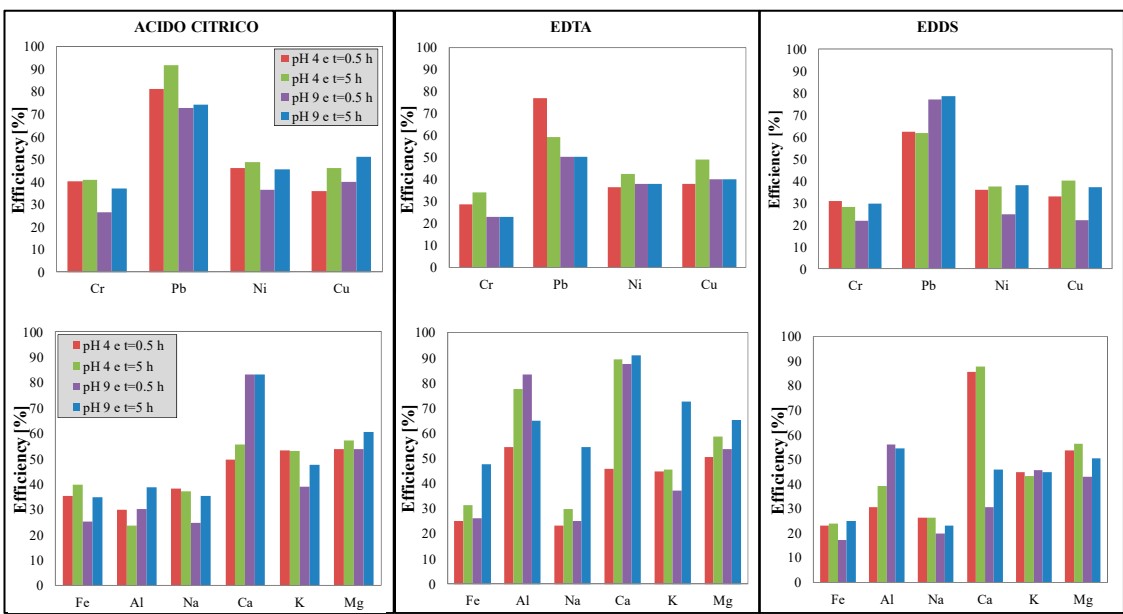

**Figure 3.** Heavy metal removal efficiency with pH-controlled during second survey campaign.

The comparison with the data reported in the previous paragraph confirms the general observations previously discussed:

- citric acid, in general, has guaranteed for the heavy metals tendentially greater removal efficiencies, even more in an acidic environment (pH 4) and a contact time of 5 h; high removal efficiencies were obtained for Pb (80%) at 0.5 h, pH 4, and citric acid;
- in the case of the use of EDTA, the removal efficiencies of inorganic contaminants showed higher values in the basic environment (pH 9) and at the contact time of 5 h, although lower than the removal efficiencies achieved with citric acid; and
- in the case of EDDS it is the agent that showed the clearest removal efficiencies except for the Pb (~78%) at pH 9 and contact time of 5 h.

Mercury deserves a separate discussion. Unlike the average behavior of other pollutants, especially inorganics, there is no evident concentration in the fine fraction of the pollutant; on the other hand, mercury has also been found in the coarse fraction [31]. This extreme variability has been charged to the different form in which mercury is bound: soluble, weakly, or strongly bound. In this context, since the problem of the variability of mercury has not been the object of in-depth research in this phase of experimentation, one can only hypothesize that:

- the different forms of mercury present in the sediments interact differently with the different granulometric sediment fractions; and
- given that the methods used for the analysis on heavy metals concern a small quantity of sample (0.5 g), it is probable that the extremely variable nature of mercury (compared to other metals) is more affected by an imperfect homogenization.

It is possible that the results have been influenced by a different previous movement of the in-situ sediment: in light of the analysis and research carried out, it is also possible to hypothesize that the soluble portion of mercury is present only when the concentrations of this in the original sample exceed a few tens of mg $kg_{DW}^{-1}$.

Finally, as regards the potential treatment by means of sediment washing, the tests carried out (all related to the first survey campaign) did not show particular and satisfactory results for Hg: the most efficient treatment was recorded with acetic acid and citric acid at a concentration of 1 M. In any case, the efficiency of removal was, at most, about 30%.

### 3.3. TPH Removal Efficiency

The assessment of potential treatment capacity of sediment washing in terms of hydrocarbon removal was performed only in the second experimental survey campaign and on the sediment fraction less than 63 μm, since it is more contaminated (17,800.002 mg $kg_{DW}^{-1}$) than the fraction above 63 μm (9500.028 mg $kg_{DW}^{-1}$). In this sense, it is necessary to underline that the analysis, in this second experimental phase, was preceded by a series of preliminary tests aimed at evaluating and validating the observations previously detected (including for heavy metals, already discussed previously). In this context, for these tests was adopted the concentration of 0.05 M and as extracting agents EDTA, citric acid, and EDDS.

With regard to the results, obtained at controlled pH, in Figure 4 the residual hydrocarbon concentrations are reported after the relative washing tests.

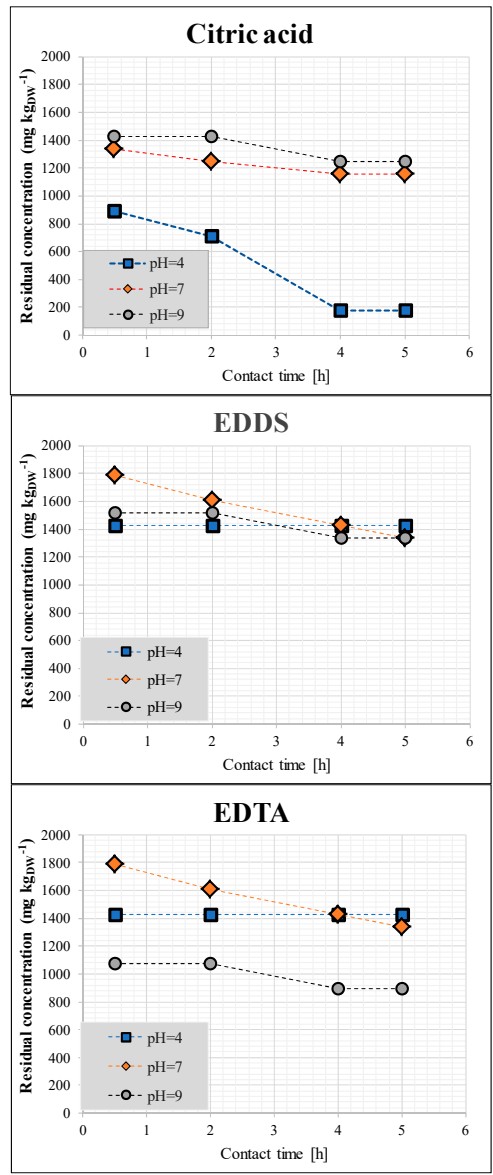

**Figure 4.** Residual TPH concentration after controlled pH washing tests.

It should be noted that the best results were obtained with citric acid at pH 4 and contact time of 4 h: in general, the basic environment favors the dissolution of the pollutants only with EDDS

and EDTA; on the other hand, citric acid seems little influenced by the alkaline environment, indeed favored by the further lowering of pH.

It is important to underline that also in this case the citric acid is satisfactory, even at neutral pH: only the batch test conducted in a strongly alkaline environment with EDTA showed an increase in performance compared to citric acid at the same boundary conditions. In this sense, a cost–benefit analysis would be useful to integrate the survey, with the aim of evaluating both the management simplification deriving from a neutral pH treatment, therefore, without pH corrections, and the economic savings linked to the use of citric acid.

Moreover, the results obtained for the removal of organic contaminants (TPH) show that for all three extracting agents analyzed, extraction efficiencies have increased with increasing contact time except in the case where EDTA was used at pH 4. At this pH value, the best extracting agent was found to be citric acid, as it achieved a removal efficiency of 99%, while EDTA showed a constant efficiency of 85% and the EDDS of 83%. At pH 7, citric acid and EDTA both achieved extraction efficiencies of about 87–88%, higher than EDDS. Finally, at pH 9, EDTA was found to be the best extracting agent achieving a 90% extraction efficiency.

### 3.4. Lay-Out of the "Sediment Washing" System

From the results obtained in the tests previously described, citric acid has been shown to be the best agent for the removal of organic and inorganic contaminants, while EDTA only for the removal of inorganic contaminants. Furthermore, based on market research, the two chelators seem to be cheaper than others: on average, citric acid costs 19 € kg$^{-1}$, compared to 82 € kg$^{-1}$ for EDTA, and even 920 € kg$^{-1}$ for EDDS.

Therefore, the choice of using citric acid as an agent for the removal of contaminants in sediments in the work appears to be the most advantageous from a technical-economic point of view.

In this sense, Figure 5 proposes a "general scheme" of train treatment, for the treatment of sediments investigated. The scheme involves washing with citric acid and the subsequent extraction of the inorganic residue with EDTA; alternatively, the EDTA could be replaced with the further dosage of citric acid at pH = 4.

The sediment washing system, proposed and schematically shown in general in Figure 5, must operate under the following conditions:

- maximum size of the workable sediments: 50 mm;
- material characteristics: loose (granular) non-cohesive/adhesive (plastic) material, with a maximum humidity of 25%, able to pass through a horizontal 90 mm light grid and allowing the use of a power supply unit equipped with extractor belt;
- final products: silts-clays < 0.063 mm, sand 0.063–2 mm, waste > 2 mm (gravel 4–25 mm, gravel 25–90 mm), organic material 0–90 mm, ferrous material 0–90 mm; and
- On the basis of these hypotheses and of the hypothesized general scheme, was assessed a variable treatment cost between 75 € m$^{-3}$ (large scale, ≈300,000 m$^3$) and 28 € m$^{-3}$ (small scale, ≈50,000 m$^3$); this estimates the amount of sediment eventually reused, was considered simply as a "savings" item for non-disposal, rather than a "gain" item for sale and valorization.

The calculation hypotheses also provided for:

- "turnkey" all-inclusive plants, with the use of "economic" agents (such as citric acid) for the treatment of both organic and inorganic compounds, with a double treatment chain in parallel for the fine fraction (90%) and coarse (10%, useful for the residual mercury and hydrocarbons);
- despite all the costs were normalized by cubic meter of the treated matrix, a series of working hypotheses were put forward which provided the scale, plant and management effects, linked above all to the capital expenditure costs, validated also on the basis of literature data;
- amortization costs have provided a "constant rate of 6%" and a useful life of the plant of 15 years; and

- the evaluation of the operating expenditure costs were made on an analytical basis, with reference to the pilot and laboratory tests carried out as part of an experimental study to be carried out.

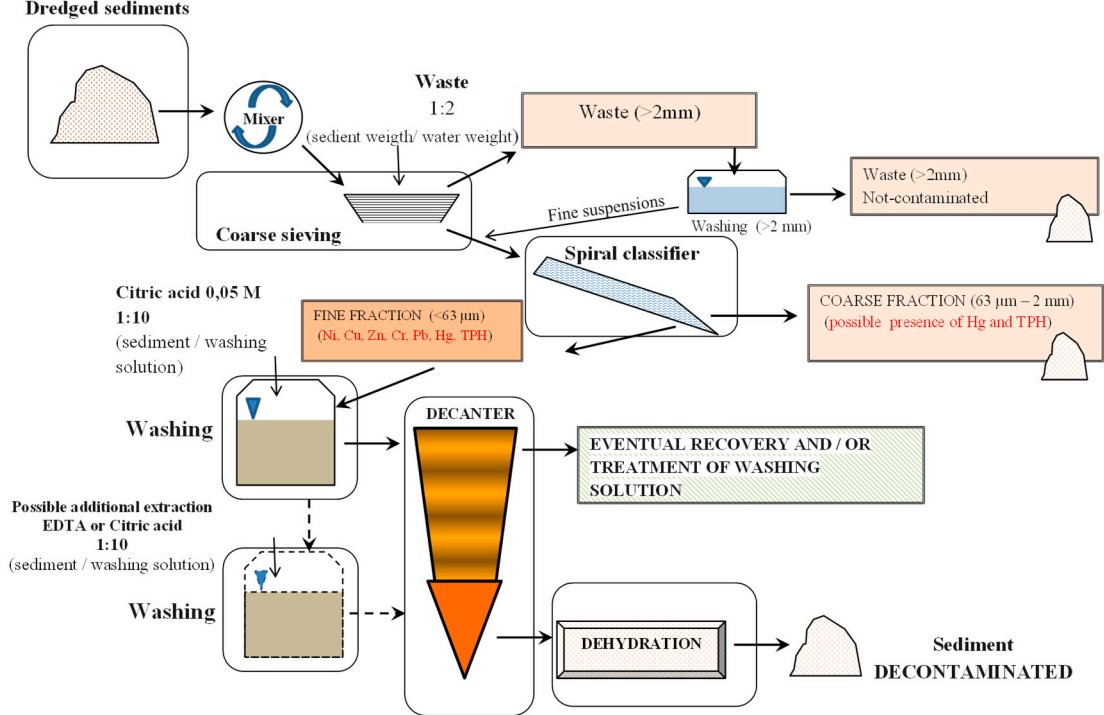

**Figure 5.** Schematization of the proposed treatment.

In summary, we could foresee:

- a small-scale pilot plant, with a total cost of 1,000,000–1,500,000 euros for the treatment of around 10–15 t h$^{-1}$;
- a medium–large-scale pilot plant, with a total cost of 1,750,000–2,500,000 euros for the treatment of approximately 25–30 t h$^{-1}$; and
- a full-scale plant, with a total cost of 5,000,000–8,000,000 euros for treatment over 60 t h$^{-1}$.

In all cases, the distribution of priority cost items includes: about 50% for galvanized machines and carpentry; about 35% for the entire waste water and mud treatment section; about 15% completion plants and services.

The total cost of remediation should, however, be compared with the management costs of these matrices, referring to the site-specific characteristics. In any case, it should be emphasized that the analysis does not take into account the "environmental benefits" and the flexibility that this type of supply chain can guarantee.

## 4. Conclusions

The characterization of the sediments has highlighted the different ability of the pollutants to interact, depending on the granulometric characteristics. This is of particular interest in the preliminary stages of assessment of the possible treatment, recovery, or disposal interventions to which the sediments are subjected. In fact, even in cases where it is possible to separate an uncontaminated coarse fraction to be initiated for recovery, it may still be useful to treat the fine fraction (for the purpose of recovery or reduction of the disposal tariff). In this sense, the characterization of the sediments "untreated" is aimed at assessing the need for intervention but is not "complete" for the purpose of the subsequent intervention and management of the "recovered sub-product", or "product waste".

From the point of view of the treatment for sediment reclamation, the washing processes based on the use of chelating agents are considered promising for the decontamination of marine sediments, as regards both organic and inorganic pollution.

The optimization of the process, of course, requires in any case a preliminary phase of analysis and depth study of many aspects: choice of the extracting agent and its concentration, influence of contact time, and pH; and solid-liquid relationship, pre-treatment, and initial particle size separation.

In the case of contaminated sediments collected in Augusta Bay, it has been shown that the reclamation can be satisfactorily carried out with sediment washing techniques, by pre-fractioning the fine fraction (<63 micrometers) and subjecting it to sequential washing (solid–liquid ratio equal to 0.1 by weight) with citric acid in neutral environment and subsequently in acidic environment or, alternatively, with citric acid first and EDTA at controlled pH, subsequently.

**Author Contributions:** The authors' individual contributions according to the relevant CRediT roles: L.L.: Data curation; formal analysis; investigation; methodology; roles/writing—original draft; M.G.G.: Data curation; formal analysis; investigation; methodology; roles/writing—original draft; G.V.: Conceptualization; formal analysis; methodology; supervision; validation; visualization; review and editing; G.D.B.: Conceptualization; formal analysis; funding acquisition; methodology; project administration; resources; supervision; validation; visualization; writing—review and editing. All authors have read and agreed to the published version of the manuscript.

**Funding:** This research received no external funding.

**Acknowledgments:** We also thank the company B.S.I. srl (2008) "Società Brescia Impianti", for estimates and support in the definition of investment costs of washing sediment.

**Conflicts of Interest:** The authors declare no conflict of interest.

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
