# Peer review of "Washing Batch Test of Contaminated Sediment: The Case of Augusta Bay (SR, Italy)"

_applsci, doi:10.3390/app10020473_

Round 1
Reviewer 1 Report
This manuscript entitled washing batch test of contaminated sediment: the case of Augusta Bay (SR, Italy) demonstrated
a detailed experimental investigation to evaluate the potential of sediment treatment of the Bay of Augusta, based on batch tests focused on use of different chelating agents and different boundary conditions (contact time, pH control, concentration of chelating agent, etc.).
The authors definitely elucidated the technical evaluation of some engineering aspects: the effect of the granulometric distribution of the sediment to be treated and the technical-economic analysis of a possible and complete layout from this study.
In general, this manuscript is suitable to "Sustainable Environmental Restoration Technologies” as a special Issue in applied sciences.
Just minor thing is please address the formats of Figures.
Figure 1 and 2 should be changed high resolution.
Figure 4, 5 and 6 should be changed in a format.
The size of fonts in Figure 7 should be changed.
Author Response
This manuscript entitled washing batch test of contaminated sediment: the case of Augusta Bay (SR, Italy) demonstrated a detailed experimental investigation to evaluate the potential of sediment treatment of the Bay of Augusta, based on batch tests focused on use of different chelating agents and different boundary conditions (contact time, pH control, concentration of chelating agent, etc.).
The authors definitely elucidated the technical evaluation of some engineering aspects: the effect of the granulometric distribution of the sediment to be treated and the technical-economic analysis of a possible and complete layout from this study.
In general, this manuscript is suitable to "Sustainable Environmental Restoration Technologies” as a special Issue in applied sciences.
Just minor thing is please address the formats of Figures.
Figure 1 and 2 should be changed high resolution.
Figures 1 and 1 have been deleted
Figure 4, 5 and 6 should be changed in a format.
The format of the Figure 4,5 and 6 have been modified.
The size of fonts in Figure 7 should be changed.
The format size of the Figure 7 have been changed as required.

Reviewer 2 Report
Dear authors,
Having read your valuable manuscript (MS), I suggest that it requires more work. Apart from language issues (grammar, and style), it has some typing errors.
My general impression is that the whole text is confusing. From the text, I can not resolve whether you presented your results, or you gave a literature review. My general suggestions are following: 1. MS should be shorter, 2. follow the topic consistently, and try not to jump from one topic to another one, 3. precisely present what exactly did you do, why, and how. Namely, you mentioned that some analyses were done previously, and it was not displayed accurately and precisely and clearly WHAT was done before, HOW, and WHY, and WHEN (citations needed!).
Abstract – since you use washing, or sediment washing, or washing batch test, try to decide which abbreviation you should use throughout the text consistently. Please, consult websites (Springer for example) with directions how to write abstract. You should write yours anew. The current form is rather loose.
Line 37 – what is SIN?
Introduction should start with line 32 (Meditterranean sea)... and try to develop the story with some logical order. What is a problem, why, and how to solve it, with reference to relevant publications. 50-61: this is very long, tiresome to follow, the style is narrative; not good. Try to present most important facts in concise way.
The problem is that you structured the text with too many paragraphs which do not relate to each other. Try to organize them so that there are few of them, according to few topics studied in your laboratory.
How many chelator agents are there? Please, focus on just few of them which exactly were used in your experimental design.
131-138: this is particularly not good. Try to point out concisely and precisely WHAT was the objective of your research.
Figures 1, 2, 3 are quite low quality; please, edit them, try to combine them into 1 or 2 figures.
ICRAM?
204-205: you have to present previous studies on your locality. What was done before, and what was done now? Try to avoid points (207-213), and do the text as text, not points.
213-220: very difficult to follow (a year later, specific points, campaign, results section, least potential, untreated... please, pay attention to clarity of the text).
So, I suggest that you introduce a table with a list of sampling campaigns, previous work, and batch experiment explanation.
223: second – what second? when? why?
249-255: quality check is missing!
262: TPH?
Table 1; units should be in miligram or microgram per kg
data in 3 significant figures should be present.
Fig 3 is low quality
Some parts (e.g. 289-297) should be moved to Introduction, or dicussion...
311-326: this part is also not very clear.
Fig 4 is very bad quality, it is difficult to resolve what is what
The rest of MS is very confusing, too much hypothethical story which is not precise and accurate interpretation of your results. I suggest you to write the whole MS anew.
Author Response
Having read your valuable manuscript (MS), I suggest that it requires more work. Apart from language issues (grammar, and style), it has some typing errors.
My general impression is that the whole text is confusing. From the text, I can not resolve whether you presented your results, or you gave a literature review. My general suggestions are following: 1. MS should be shorter, 2. follow the topic consistently, and try not to jump from one topic to another one, 3. precisely present what exactly did you do, why, and how. Namely, you mentioned that some analyses were done previously, and it was not displayed accurately and precisely and clearly WHAT was done before, HOW, and WHY, and WHEN (citations needed!).
Abstract – since you use washing, or sediment washing, or washing batch test, try to decide which abbreviation you should use throughout the text consistently. Please, consult websites (Springer for example) with directions how to write abstract. You should write yours anew. The current form is rather loose.
The abstract has been rewritten.
Line 37 – what is SIN?
The authors have included a brief description.
Introduction should start with line 32 (Meditterranean sea)... and try to develop the story with some logical order. What is a problem, why, and how to solve it, with reference to relevant publications. Corrections have been made.
50-61: this is very long, tiresome to follow, the style is narrative; not good. Try to present most important facts in concise way.
The period 50-61 has been reduced.
The problem is that you structured the text with too many paragraphs which do not relate to each other. Try to organize them so that there are few of them, according to few topics studied in your laboratory.
The authors modified the paragraphs concerning the description of the tests to clarify what was done in the laboratory.
How many chelator agents are there? Please, focus on just few of them which exactly were used in your experimental design.
The authors modified paragraph 2.2, describing in detail the tests performed and the types of chelating agents used.
131-138: this is particularly not good. Try to point out concisely and precisely WHAT was the objective of your research.
The authors have rewritten the period 131-138.
Figures 1, 2, 3 are quite low quality; please, edit them, try to combine them into 1 or 2 figures.
Figures 1 and 2 have been deleted and Figure 3 has been modified.
ICRAM?
The authors modified the paragraph containing the characterization of sediments carried out by the ICRAM to give greater clarity to the manuscript.
204-205: you have to present previous studies on your locality. What was done before, and what was done now? Try to avoid points (207-213), and do the text as text, not points.
The paragraph has been modified and the period (207-2013) has been reported as text and not with points
213-220: very difficult to follow (a year later, specific points, campaign, results section, least potential, untreated... please, pay attention to clarity of the text).
The corrections have been made by modifying the paragraph 2.2.
223: second – what second? when? why?
The paragraph has been corrected specifying that the work was carried out with marine sediments taken in two different experimental campaigns and in different points of the Augusta Bay.
249-255: quality check is missing!
249-225:It has been modified specifying the international reference methods.
262: TPH?
In the text, it has been specified that the acronym TPH means the Total Ptroleum Hydrocarbons, measured from C12 to C40.
Table 1; units should be in miligram or microgram per kg.
Data were expressed in miligram per kg.
Data in 3 significant figures should be present.
Data were expressed with 3 figures significant.
Fig 3 is low quality.
Fig 3 has been corrected.
Some parts (e.g. 289-297) should be moved to Introduction, or discussion.
The part (289-297) has been moved to the introduction.
311-326: this part is also not very clear.
This part has been corrected.
Fig 4 is very bad quality, it is difficult to resolve what is what.
Fig 4 has been modified.
The rest of MS is very confusing, too much hypothetical story which is not precise and accurate interpretation of your results. I suggest you to write the whole MS a new.
Corrections were made in the remaining part of the manuscript, in particular in the interpretation of the results (paragraphs 3.2 and 3.3).
